# Chronological Age Affects MSC Senescence In Vitro—A Systematic Review

**DOI:** 10.3390/ijms22157945

**Published:** 2021-07-26

**Authors:** Konstantinos Kapetanos, Dimitrios Asimakopoulos, Neophytos Christodoulou, Antonia Vogt, Wasim Khan

**Affiliations:** 1School of Clinical Medicine, University of Cambridge, Cambridge CB2 2SP, UK; da464@cam.ac.uk (D.A.); nc508@cam.ac.uk (N.C.); 2Division of Trauma & Orthopaedic Surgery, Addenbrooke’s Hospital, University of Cambridge, Cambridge CB2 0QQ, UK; av591@cam.ac.uk (A.V.); wasimkhan1@nhs.net (W.K.)

**Keywords:** mesenchymal stromal cells, senescence, ageing, biomarkers, systematic review, human

## Abstract

The use of mesenchymal stromal cells (MSCs) in regenerative medicine and tissue engineering is well established, given their properties of self-renewal and differentiation. However, several studies have shown that these properties diminish with age, and understanding the pathways involved are important to provide regenerative therapies in an ageing population. In this PRISMA systematic review, we investigated the effects of chronological donor ageing on the senescence of MSCs. We identified 3023 studies after searching four databases including PubMed, Web of Science, Cochrane, and Medline. Nine studies met the inclusion and exclusion criteria and were included in the final analyses. These studies showed an increase in the expression of p21, p53, p16, ROS, and NF-κB with chronological age. This implies an activated DNA damage response (DDR), as well as increased levels of stress and inflammation in the MSCs of older donors. Additionally, highlighting the effects of an activated DDR in cells from older donors, a decrease in the expression of proliferative markers including Ki67, MAPK pathway elements, and Wnt/β-catenin pathway elements was observed. Furthermore, we found an increase in the levels of SA-β-galactosidase, a specific marker of cellular senescence. Together, these findings support an association between chronological age and MSC senescence. The precise threshold for chronological age where the reported changes become significant is yet to be defined and should form the basis for further scientific investigations. The outcomes of this review should direct further investigations into reversing the biological effects of chronological age on the MSC senescence phenotype.

## 1. Introduction

Mesenchymal stromal cells (MSCs) are a heterogeneous population of stromal cells capable of self-renewal and multi-lineage differentiation into various tissues of mesodermal origin [1]. Although MSCs were originally isolated from bone marrow [2], recent research has identified alternative sources, including adipose tissue [3], dental pulp [4], the endometrium [5], peripheral blood [6], the periodontal ligament [7], the placenta [8], the synovial membrane [9], and umbilical cord blood [10]. Evidence has suggested that MSCs can be found in all vascularized tissues of the body [11]. MSCs have a high proliferative potential, multipotency, immunomodulatory activity, and paracrine effect [12,13,14]. These features, as well as the lack of ethical concerns that surround embryonic stem cells, make MSCs a promising material for cell therapy, regenerative medicine, immune modulation, and tissue engineering [15]. In the last decade, they have been increasingly used in clinical practice to treat numerous traumatic and degenerative disorders [16].

Although there is no consensus on a single surface molecule to identify MSCs, the International Society for Cellular Therapy (ISCT) proposed a minimum of three criteria that need to be satisfied for a cell to be considered an MSC [17]: (a) adherence to plastic under standard culture conditions; (b) expression of CD73, CD90, and CD105, and lack of expression of CD11b or CD14, CD34, CD45, CD19 or CD79a, and HLA-DR; (c) differentiation into adipocytes, chondrocytes, and osteoblasts in vitro. Other surface markers generally expressed by MSCs include CD10, CD13, CD29, and CD44 [18,19].

Although MSCs are present in several tissues, their numbers are low within the aspirate. Thus, to be clinically useful, MSCs need to be expanded in vitro over several population doublings (PDs) to obtain enough cells before implantation. The chronological age of a donor strongly influences the quality and lifespan of MSCs [20,21]; MSCs from aged donors perform less well than their younger counterparts because of their reduced proliferative capacity and differentiation potential [22]. Additionally, their regenerative potential declines after 30 years of age [23]. This is further supported by the observation that human umbilical cord MSCs, being neonatal in origin, can grow up to passage 10 without losing multipotential capacity [24], and exhibit enhanced expression of telomerase and pluripotency factors compared to adult bone marrow-derived MSCs [25].

While in vivo ageing refers to the chronological age of the donor, which affects the lifespan of MSCs, in vitro ageing is the loss of MSC characteristics as they acquire a senescent phenotype during expansion in culture. Irrespective of donor age, during in vitro expansion, the proliferation rate progressively decreases until growth arrests [26,27,28,29]. This was first proposed by Hayflick in 1965, who suggested that all primary cells cease to proliferate after a certain number of cell divisions in a process known as cellular senescence [30]. Senescence negatively affects the differentiation capacities of MSCs, resulting in reduced efficacy following transplantation [31]. Several biological markers have been proposed to drive senescence. These include p16, p21, p53, and nuclear factor kappa B (NF-κB) [32,33]. They take part in several biochemical pathways of the cell cycle. Other markers, such as superoxide dismutase (SOD), are found to inhibit the cell from reaching that state [34]. The levels of the aforementioned markers differ as the MSCs undergo chronological ageing.

For MSCs to be clinically effective in an ageing population, it would be of great significance to monitor senescence and understand the molecular basis of chronological ageing. This systematic review aims to provide the reader with the current knowledge about the effect of chronological ageing on MSC senescence. The molecular mechanisms involved in the age-related decline of the therapeutic potential of MSCs will also be discussed.

## 2. Results

### 2.1. Characteristics of Included Studies

Nine studies were included in this systematic review, as per Table 1. Sample sizes ranged from 8 to 57, while the donors’ chronological age ranged from 6 to 92 years of age. All studies used a system of dividing donors into age groups, and only data for patients over 18 years of age were analysed. These data are shown in detail in Table 1. Four studies used adipose tissue for the purposes of their experiments and three studies used bone marrow. Pandey et al. (2011) [33] used both adipose- and bone marrow-derived MSCs, while Ferretti et al. (2015) [35] used periosteal tissue.

### 2.2. Cell Surface Characterisation

In terms of cell surface characterisation, all studies analysed the cell surface markers of the cells, with all but Ferretti et al. (2015) [35] presenting their results. Eight studies looked at CD90, seven studies looked at CD105, and five studies looked at CD73. Although it appears that CD90 is a universally accepted marker for the definition of MSCs, the rest of the surface markers proposed by ISCT were not always used. Most of the included studies showed CD34- [32,33,36,37,38,39,40] and CD45- [32,33,36,37,38,39] cell surface characterisation, as per the criteria of ICST. Several surface markers not included in the ICST criteria are repeatedly seen in the included studies as defining features of an MSC population including CD44 [32,33,34,36,37,38,39,40], CD29 [33,37,38,40], CD146 [36,37], and CD166 [33,37,38].

### 2.3. Molecular Markers of Senescence

The markers of senescence are outlined in Table 2. Stolzing et al. (2008) [34] measured senescence by means of flow cytometry of p21 and p53; reactive oxygen species (ROS); nitric oxide (NO); advanced glycation end-products (AGEs), and their receptors (RAGEs); SOD; heat shock proteins HSP27, HSP70, and HSP90; as well as by measuring the rate of apoptosis. Wagner et al. (2009) [37] assessed the upregulation or downregulation of various genes. Pandey et al. (2011) [33] measured NF-κB in adipose MSCs (ASCs), as well as the proteomic, miRNA, and RNA profile. Alt et al. (2012) [36] looked at CHEK1, p16ink4a, ATM, E2F4 transcription factor, Rb1, BRCA1, HMGA2, ATR, NFκB, TNFa, hTERT, p53, Casp3, Casp8, Casp9, XRCC4, XRCC6, APEXZ1, Let7g, mir-27B, mir-106a, and mir-199a. Siegel et al. (2013) [38] looked at Oct4, Nanog, Prdm14, and SOX2 as markers of pluripotency. Choudhery et al. (2014) [39] measured the levels of p16, p21, SA-β-galactosidase, and SOD. Ferretti et al. (2015) [35] measured Ki67, p53, NO, Bmp2, Runx2, IL-6, OPG, RANKL, Oct4, Nanog, and Sox2. Marędziak et al. (2016) [40] measured SA-β-galactosidase, ROS, NO, SOD, p21, p53, and the number of dead cells. Finally, Liu et al. (2017) [32] focused on SA-β-galactosidase, mitochondrial superoxide content, ROS, the activity of the proteasome, hTERT1, and SIRT1, as well as p16, p21, and p53.

### 2.4. Increased Expression of p21, p53, p16, ROS, and NF-κB with Age

p21 and p53 were assessed with flow cytometry and qRT-PCR and showed a significant increase in expression with chronological age, with Stoltzing et al. (2008) [34] reporting a 3-fold and 6-fold increase when comparing adult with young and aged with young groups, respectively. Choudhery et al. (2014) [39], Marędziak et al. (2016) [40], and Liu et al. (2017) [32] further reported an increase in the expression of p21 with age, while Ferretti et al. (2015) [35], Marędziak et al. (2016) [40], and Liu et al. (2017) [32] reported an increase of p53 expression levels with age. Only Alt et al. (2012) [36] reported downregulation of p53, as measured through real-time PCR (RT-PCR). With regards to p16, Alt et al. (2012) [36], Choudhery et al. (2014) [39], and Liu et al. (2017) [32] reported a significant increase in its expression levels with age, as measured with quantitative RT-PCR, with Alt et al. reporting a 22.9-fold increase.

Levels of ROS were found to significantly increase with age, as reported by three studies [32,34,40]. Results were mixed with regards to the expression levels of NO and SA-β-galactosidase. With respect to SA-β-galactosidase, two studies reported an increase in its levels with age [32,39], and one reported no statistically significant differences [40]. Two studies concluded that the levels of apoptotic regulator NF-κB increase with age, through the use of immunohistochemistry, Western blot analysis, and RT-PCR [33,36]. Pandey et al. (2011) investigated the changes of NF-κB through an immunohistochemical, proteomic, and RNA profile, which showed an increase in its expression with age. Although Pandey’s RNA profile showed an increase in the expression of non-canonical targets of NF-κΒ, they also reported a decrease in the expression of canonical targets [33].

### 2.5. Decreased Expression of SOD with Age

Stolzing et al. (2008) [34], Choudhery et al. (2014) [39], and Marędziak et al. (2016) [40] reported a decrease in the expression levels of SOD with age, as measured through the use of assay kits.

## 3. Discussion

While in vivo ageing refers to the chronological age of the donor, in vitro ageing is the loss of MSC characteristics as they acquire a senescent phenotype during in vitro expansion. For MSCs to be clinically effective in an ageing population, it would be of great significance to monitor senescence and understand the molecular basis of chronological ageing.

The diverse nature of MSCs poses a great challenge to forming a precise characterisation profile [41]. The minimal criteria to define MSCs, as proposed by the International Society for Cellular Therapy (ISCT) [17], were an important step towards the standardisation of MSC models. However, it is yet unclear whether these criteria are accurate, essential, and comprehensive. Several of the studies included in this review described an MSC population that included some, but not all, of the minimal criteria [32,33,34,36,37,38,39,40]. The heterogeneity between MSC populations in different studies prevents the accurate comparison of outcomes. This poses a particular challenge to the advancement of the field. A potential solution would be to rely on functional assays, as well as surface marker characterisation, to accurately define an MSC population. This could involve differentiation potentials [17] and immunological characterisation [42].

### 3.1. Senescence

Senescence is a concept with an elusive definition. Broadly, it describes cellular deterioration due to ageing, and a subsequent loss of growth and proliferative capacity. Senescence is a well-documented outcome in cellular populations that undergo genotoxic stress [43]. This is reflected in a cell’s absence of proliferation, the expression of senescence markers, and the secretion of a senescence-associated secretory phenotype (SASP). It is an irreversible state that protects a cell from the consequences of compromised genomic integrity [44]. However, there is no single phenotypic definition to describe the process of senescence, but rather a collection of phenotypes. Specifically, hallmarks of cellular senescence include an arrest of the cell cycle, resistance to apoptosis, SA-β-gal expression, and a SASP that is associated with growth-related oncogene (GRO), IL-8, IL-12, and macrophage-derived chemokine (MDC) [45]. The driving cause of senescence with ageing appears to be telomere dysfunction, as well as associated genetic and epigenetic changes, perceived as DNA damage and activating the persistent DNA damage response (pDDR) mechanism [46,47]. Human MSCs have been described as exhibiting a marked resistance to apoptosis and to default to a senescence phenotype in response to injury [31]. When exposed to ionising radiation, p53 and p21 were seen to drive MSC cycle arrest and p16 and RB to drive senescence [48]. Similar markers were explored by some of the studies included in this review, with SA-β-gal, inflammatory markers, and pDDR proteins being most frequently referenced [32,33,34,36,39]. A reduction in the rates of proliferation, as well as an increase in the senescence phenotype was observed in similar experiments that evaluate the effects of in vitro ageing on MSCs [49].

### 3.2. Cell Cycle Arrest and Apoptosis

When assessing the effects of chronological age on MSC senescence it is essential to evaluate evidence for cell cycle arrest and against proliferation. In the included studies, p21 and p53 were seen to be significantly upregulated in six studies [32,34,35,36,39,40], providing strong evidence for cell cycle arrest, as both molecules are central components of the response pathway to DNA damage [50]. In addition, upregulation of p16 [32,36,39], ATM [36], CHK1 [36], E2F4 [36], and RB [36] (illustrated in Figure 1, below) was also noted in MSC samples from older donors, highlighting the persistent activation of the DNA damage response pathway [51]. Although the upregulation of all mentioned markers was rarely seen in a single study, the six studies mentioned the upregulation of at least one marker, with the remaining three not mentioning whether they looked for a change in their expression. Furthermore, highlighting the effects of an activated pDDR in cells from older donors, a decrease in the expression of proliferative markers like Ki67, MAPK pathway (illustrated in Figure 1, below) elements, and Wnt/β-catenin pathway elements was seen in two studies [33,35]. It is unclear, however, whether the absence of proliferation was associated with an increase in the rate of apoptosis. It is known that MSCs exhibit resistance to apoptosis [17], and so do most senescent cells, but the evidence in the literature was contradicting. Specifically, both mention that an increase in the rate of apoptosis [34] and a decrease in the expression of caspases 3/8/9 [36] was seen in two different studies. Two studies reported an increase in the expression of SA-β-galactosidase, supporting the association between senescence and chronological age [32,40].

### 3.3. Stress and Inflammation

In line with the evidence provided for the activation of stress-response pathways, there was evidence of an increase in stressful agents in older donors. An increase in the quantified levels of ROS [32,34,40], NO [34,35,40], and advanced glycosylation end-products and their receptors (AGEs and RAGEs) [34] was reported, as well as a decrease in the levels of SOD [34,39,40]. In line with an increase in markers of stress, there was an increase in the expression of inflammatory markers amongst the older donor group. Specifically, there was a reported increase in NF-κB [33,36], as well as TNFα [36] and IL-6 [35]. A potential approach to subsequent studies could be the investigation of any functional consequences on the immunological environment of MSCs from older donors, with over-expressed inflammatory markers. Lastly, there is contradicting evidence on the expression of potency markers in MSCs from older donors. Siegel et al. 2013 provided evidence that there is no difference in the expression of Oct4, Sox2, or Nanog between the two groups [38], however, Ferretti et al. (2015) suggest an increase in the expression of Nanog and Sox2, and a decrease in Oct4, in the group of older donor MSCs [35]. Further investigations into the effect of chronological age and MSC potency are crucial to provide a deeper understanding of a potential mechanism of age-induced senescence in MSCs.

### 3.4. Strengths and Limitations

The systematic approach to constructing a research question yielded a thorough review of the literature to identify all relevant studies. The ability to make effective comparisons between the studies was, however, limited by the heterogeneity of the included studies in defining the MSC population; not all studies followed the minimal defining criteria proposed by ISCT to define their cell population. There was a large variation in the markers explored by the different studies with most studies not investigating the effect of chronological age on the SASP or the differentiation potential of MSC, both defined hallmarks of senescence [45]. The included genetic studies offered a descriptive assessment of genes that were up- and downregulated without any implications on how these genes affect senescence pathways. Further work is needed to identify the relevant pathways through which chronological age influences senescence in MSCs.

## 4. Materials and Methods

### 4.1. Databases

A systematic review of the existing literature was conducted, as per the guidelines of the Preferred Reporting Items for Systematic Reviews and Meta-Analyses checklist [52]. The Prospero Registration Number of this systematic review is 259531. Four databases were used for the literature search; PubMed, Cochrane, Web of Science, and Medline.

### 4.2. Search Terms and Inclusion and Exclusion Criteria

Our search strategy involved terms of “age” or “ageing” and “mesenchymal stem cells” or “mesenchymal stem cell” or ”mesenchymal stromal cells” or “mesenchymal stromal cell” and “cell surface characterisation” or “cell surface” or “differentiation potential” or “differentiation” and “in vitro”.

Inclusion of a study required human subjects and reference to their chronological age. Only studies referring to MSCs, as opposed to other sources of cells, were included, while there should be a specific reference for the extraction of the MSCs from a particular tissue source. Included studies had to include the proliferation, characterisation, and differentiation of MSCs to allow for assessment adherence to ISCT criteria. Only studies looking at in vitro models were included and had to be written in the English language. Exclusion criteria included animal models and in vivo models. All case studies, editorials, commentaries, reviews, and letters to the editor (as opposed to primary research) were excluded. The above inclusion and exclusion criteria are an exhaustive list of limitations for the review. The search strategy and inclusion and exclusion criteria were finalised by AV and WK, and they resolved all potential disputes.

### 4.3. Data Collection and Extraction

In week 4 of January and week 1 of February 2021, DA, NC, and KK independently conducted a search of the existing literature on the topic, using the four aforementioned databases and extracted 988 studies from PubMed (1996—week 4 of January 2021), 23 studies from Cochrane (1946—week 4 of January 2021), 1309 studies from Web of Science (1900—week 4 of January 2021), and 703 studies from Medline (1946—week 1 of February 2021). A total of 966 duplicates were removed, and the rest of the studies were screened and excluded with reasons. Then 144 full-text articles were assessed and a final list of 9 were used for the purposes of data extraction, following a 2-step screening process, as explained in detail in Figure 2. The process was supervised by AV and WK.

The data from each included study were extracted and organised on an Excel spreadsheet, listing the name of the primary author, publication year, brief description of the study, number of subjects and chronological age, source of the MSCs, culture conditions, proliferation analysis and results, MSC cell surface characterisation, protocols, and results for chondrogenic, adipogenic, osteogenic, and other reported outcomes. AV and WK evaluated and finalised the format of the data.

### 4.4. Quality Assessment of Included Studies

Each of the included studies was appraised independently by DA, NC, and KK, using the Office of Health Assessment and Translation tool (OHAT tool) [54], by answering 11 questions assessing the risk of bias of each study. A consensus with discussion was reached regarding each study. Results of the OHAT analysis are included in the Appendix A.

## 5. Conclusions

Our systematic review supports an association between chronological age and MSC senescence, with chronological ageing associated with an increased expression of the DNA damage response proteins, markers of ageing, stress, and inflammation. We were not able to identify a threshold of chronological age after which the phenotypic features of senescence begin to appear. The outcomes of this review should direct further investigations into reversing the biological effects of chronological age on the MSC senescence phenotype. Whilst doing so, it is essential that a clear definition of MSC characterisation is maintained across studies, ideally, aligned with the definitions suggested by the ISCT [17], and a clear set of senescence features are investigated, like markers of cell cycle arrest, apoptosis, stress, inflammation, differentiation, and SASP. This would help to develop a better understanding of the pathways involved and facilitate translation to clinical practice.

## Figures and Tables

**Figure 1 ijms-22-07945-f001:**
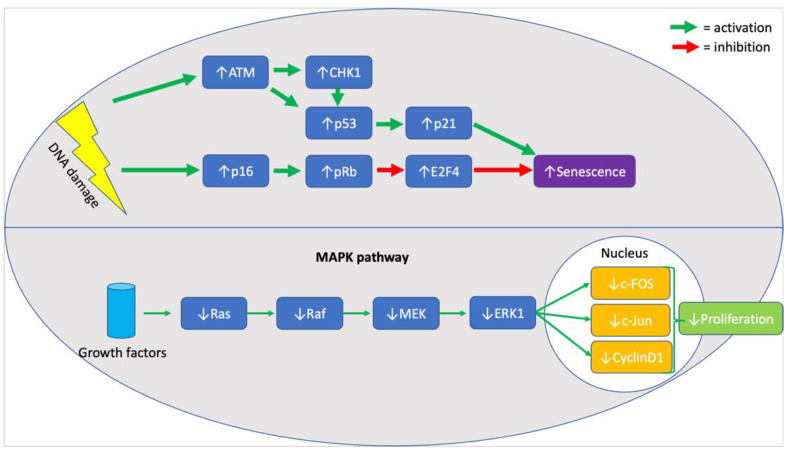
Summary of molecular markers implicated in the senescence phenotype of aged MSCs.

**Figure 2 ijms-22-07945-f002:**
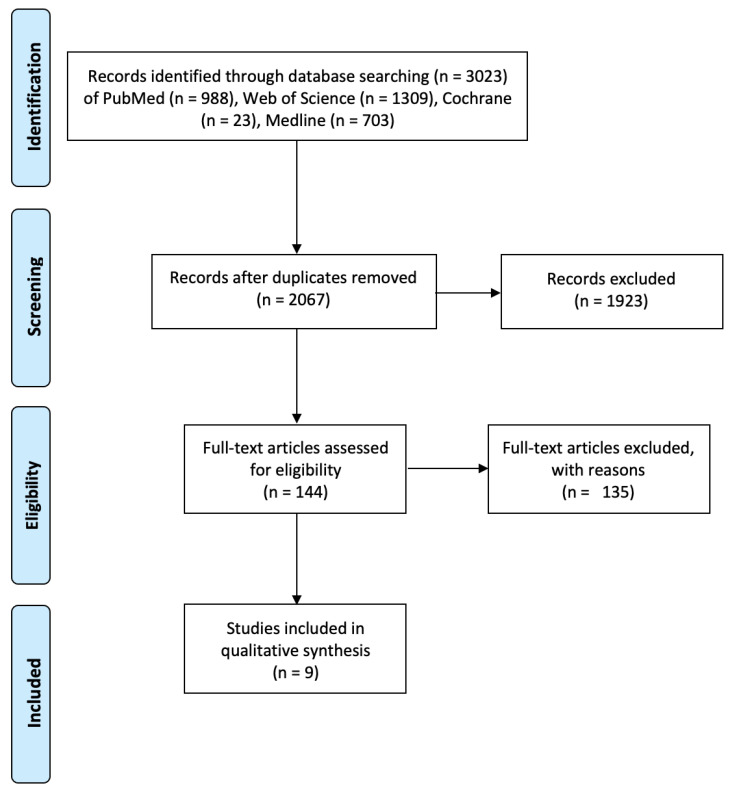
Overview of the identification, screening, and eligibility of studies, using the PRISMA Flow Diagram [53].

**Table 1 ijms-22-07945-t001:** Key characteristics of the included studies.

Primary Author (Publication Year)	Brief Description	Source	Sample Size	Donor Ages (All Numbers in y.o.)	Characterisation
Stoltzing et al. (2008)	Assessed cellular “fitness” by means of proliferation rates and markers of cellular ageing	BM from posterior iliac crest and purchased (frozen)	57	Young (7–18) Adult (19–40) Aged (41–55)	CD13+ CD44+ CD90+ CD105+ Stro-1+ D7-Fib+ Irrespective of age Levels of CD44, CD90, CD105, and Stro-1 showed significant age-related changes
Wagner et al. (2009)	Compared the genetic profiles of younger vs. older donors and pinpointed 67 age-induced and 60 age-inhibited genes	BM aspirates for allogeneic transplantation or femoral head	12	Young (21–25) Adult (44–55) Elder (80–92)	CD13+ CD29+ CD44+ CD73+ CD90+ CD105+ CD146+ CD166+ CD31- CD34- CD45- No age-related difference
Pandey et al. (2011)	Investigated changes in miRNA expression of donor ASCs and BMSCs	Subcutaneous white adipose tissue and BM from iliac crest and marrow discarded during orthopaedic procedures.	16	Young ASCs (31.5 ± 10.4) Old ASCs (63 ± 6) Young BMSC (31.5 ± 8.7) Old BMSC (56.3 ± 5)	CD29+ CD44+ CD90+ CD105+ CD166+ HLA-I+ CD3- CD34- CD45- CD11b- CD19-
Alt et al. (2012)	Investigated the repair and regeneration potential and molecular features of ASCs; found increase of CHEK1 and p16ink4a with age and abnormal levels of mir-27b, mir-106a, mir-199a, and let-7.	Abdominal Adipose Tissue	40	Young (<20, mean 16.75 ± 1.4) Middle (30–40, mean 34.4 ± 1.6) Old (>50, mean 61.33 ± 7.4)	CD44+ CD90+ CD105+ CD146+ CD3- CD4- CD11b- CD34- CD45-
Siegel et al. (2013)	Investigated effects of age on markers of senescence and compared their levels in BMSCs and pluripotent MSCs	BM during orthopaedic procedures, from patients without metabolic or neoplastic diseases	53	13–80	CD29+ CD44+ CD59+ CD73+ CD90+ CD105+ CD140b+ CD166+ HLA-ABC+ in >96% of cells ≤0.5% cells being CD14+ CD19+ CD34+ CD43+ CD45+ CD86+ CD93+ CD133+ CD243+ SSEA-1+ Another group of cells with big heterogeneity, ranging from 2.6% to 84.2% Negative correlation of CD71+ CD146+ CD274+ with increasing age and females to males
Choudhery et al. (2014)	Investigated the effect of age on population doublings, SOD activity, differentiation, and senescence	Adipose Tissue from liposuction procedures under local anaesthesia	11	<40 years of age > 50 years of age	CD44+ CD73+ CD90+ CD105+ CD3- CD14- CD19- CD34- CD45-
Ferretti et al. (2015)	Investigated the effect of age on markers of periosteum-derived precursor cells, including Ki67, p53, and NO	Periosteal tissue during surgery for orthopaedic trauma	8	2 with mean age 16 2 with mean age 28 2 with mean age 63 2 with mean age 92	N/A
Marędziak et al. (2016)	Investigated the age-dependent increase in hASC senescence	Subcutaneous adipose tissue during total hip/knee arthroplasty or other open procedures connected with fracture reduction and fixation	28	Group 1 > 20 Group 2 > 50 Group 3 > 60 Group 4 > 70	CD90+ CD73b+ CD44+ CD29+ CD34-
Liu et al. (2017)	Investigated the effect of ageing on various biomarkers and genetic markers, including SA-β-galactosidase, ROS, proteasome activity, hTERT1, SIRT1, p53, p21, p16, RB1 & RB2	Subcutaneous adipose tissue from right chest regions during various surgical procedures	24	Children (6 to 12) Young adult (22 to 27) Elderly (60 to 73)	CD44+ CD73+ CD90+ CD105+ CD34- CD11b- CD19- CD45- HLA-DR- No significant difference between groups

**Table 2 ijms-22-07945-t002:** Markers of molecular senescence in the included studies.

Primary Author (Publication Year)	Molecular Markers of Senescence Measured	Means of Measurement	Outcomes	Other Points
Stolzing et al. (2008)	p21 and p53	Flow cytometry	Age-related increase (adult = 3x young, aged = 6x young)	Age-related reduction in cell numbers and differentiation potential were observed.
	ROS		Significantly increased in aged	
	NO		Age-related increase (adult = 3x young, aged = 6x young)	
	Advanced glycation end-product (AGE), Receptors for AGEs (RAGE), Carbonyls		Age-related increase	
	SOD	Using specialised kit	Age-related decrease (aged = 0.6x young)	
	Heat Shock Proteins (HSP27, HSP70 and HSP90)	Immunohistochemistry	Age-related decrease	
	Rate of apoptosis		Age-related increase	
Wagner et al. (2009)	Numerous genes assessed for upregulation or downregulation	PCR, upregulation, or downregulation with *p*-value < 0.01	67 genes upregulated, including MEOX2, SHOX2, HOXC6 60 genes downregulated, including HOXA5, HOXB3, HOXB7, PITX2	Markers of chronological senescence showed similar changes to markers of replicative senescence.
Pandey et al. (2011)	NF-κB in adipose derived MSCs (ASCs)	Immunohistochemistry	Young: NF-κB localised mostly in nucleus Old: NF-κB localised in the nucleus too, but elevated levels were found in the cytoplasm.	Differential expression of miRNA is an integral component of biologic ageing in MSCs.
	Proteomic profile	Western blot analysis	Levels of p-IκB, p-Iκk, iNOS, ERK1/2, phosphorylated c-fos, c-jun and JNK were significantly decreased in ASCs from older donors. NF-κB/p65 levels were significantly elevated in ASCs from older donors	
	miRNA profile	qPCR	BMSCs: 45 miRNA molecules were different in expression, of which 43 were downregulated and 2 were upregulated. ASCs: 14 miRNA molecules were different in expression, of which 12 were downregulated and 2 were upregulated.	BMSCs: Increased activity found in PTEN, mTOR, RAN pathways with age, and decreased activity in Wnt/β-cat, tight-junction, SAPK/JNK signalling, cleavage, and polyadenylation of pre-mRNA ASCs: Increased activity in ephrin receptor signalling and PPARα/RXRα with age, and decreased activity in RAN and AMPK signalling and cell-cycle regulation.
	RNA profile	(Real-time) RT-PCR	Increased NF-κB and non-canonical targets IL-4 receptor and myc in old. Also increased WNT/β-cat in old. Decreased MAPK elements, iNOS, VCAM1, and IKK, and NF-kB downstream canonical targets. Decreased cell-cycle regulators such as cyclin-dependent kinases.	
Alt et al. (2012)	CHEK1 (G2 phase arrest)	RT-PCR	4.4-fold increase (old compared to young)	Age-related decline of multi-lineage differentiation potential in MSCs Adipogenic differentiation was downregulated from about 33% in Group 1 to about 10% in Group 3 Osteogenic differentiation was downregulated from about 50% in Group 1 to about 22% in Group 3
	CDK inhibitor p16ink4a (G2 phase arrest)		22.9-fold increase	
	ATM, E2F4 transcription factor, Retinoblastoma 1 (Rb 1), BRCA1, HMGA2	RT-PCR	Marginally increased	
	ATR apoptotic regulator	RT-PCR	2.6 ± 0.9-fold increase in group 3 vs. group 1	
	NFkB apoptotic regulator		4 ± 0.4-fold increase	
	TNFa apoptotic regulator		5 ± 1.4-fold increase	
	hTERT, p53, Casp3, Casp8, Casp9		Decreased	
	XRCC4 (cellular DNA repair)		3 ± 1.1-fold increase	
	XRCC6 (cellular DNA repair)		4.8 ± 0.4-fold increase	
	APEX1 (cellular DNA repair)		1.15 ± 0.4-fold decrease	
	Let7g	qRT-PCR	3.95 ± 0.2-fold decrease	
	mir-27B		3.43 ± 0.7-fold decrease	
	mir-106a		0.94 ± 0.3-fold decrease	
	mir-199a		4.68 ± 0.6-fold decrease	
Siegel et al. (2013)	Oct4, Nanog, Prdm14 and SOX2 as markers of pluripotency	qRT-PCR	No correlation with donor age	Prdm14 mRNA expression is positively correlated to the clonogenic potential of MSCs in vitro
Choudhery et al. (2014)	p16	qRT-PCR	Significantly higher in >50 y.o. than in <40 y.o.	Adipogenic differentiation potential of AT-MSCs is independent of donor age, osteogenic and chondrogenic potential decrease AT-MSCs undergo a neuronal-like differentiation irrespective of donor age
	p21			
	SA-β-gal	SA-β-gal staining kit	12.2% ± 1.1% in >50 years vs. 5.2% ± 1.9% in <40 years	
	SOD	commercially available colorimetric assay kit	26.0 ± 2.3 in <40 years of age vs. 11.7 ± 2.9 in >50 years of age	
Ferretti et al. (2015)	Ki67	Immunohistochemistry	Lowest in 92 y.o. and highest in 16 y.o.	N/A
	p53		Increase in 92 y.o. compared to 28 y.o. and 16 y.o.	
	NO	NO production	Increase in 92 y.o. compared to 28 y.o. and 16 y.o.	
	Bmp2	qRT-PCR	No differences in 16 and 28 y.o. Increase in 63 and 92 y.o. with the highest value in 63 y.o.	
	Runx2		Significantly increased in 16 and 92 y.o. compared to 28 and 63 y.o.	
	IL-6		Significant increase only in 92 y.o.	
	OPG		Increased in both 16 and 92 y.o. compared to 28 and 63 y.o.	
	RANKL			
	Oct4		Significantly decreased in 16 y.o. compared to the rest	
	Nanog		Increased in both 28 and 92 y.o. cells	
	Sox2			
Marędziak et al. (2016)	SA-β-galactosidase	Senescence Cells Histochemical Staining Kit	Statistically insignificant differences	N/A
	ROS level	Measurement of H2DCF-DA	Significantly decreased in young patients (>20) No significant differences between older age groups	
	NO levels	Griess reagent kit	Increased with age	
	SOD	SOD Assay kit	Decreased with age	
	Dead cells	Cell stain Double Staining Kit	Increased with age, but not with statistical significance	
	P21 and P53	N/A	Increased in older age groups	
Liu et al. (2017)	SA-β-galactosidase	β-galactosidase Staining Kit	Significant increase in passage 5 cells in elderly group compared to child group	N/A
	Mitochondrial superoxide content	MitoSOX Red reagent	Increased in elderly group	
	Total ROS	DCFH-DA	Increase with age	
	Proteasome activity	Proteasome activity assay kit	Decreasing trend in young adult group and elderly group compared to child group	
	hTERT1 and SIRT1	qRT-PCR	Statistically insignificant differences	
	p53, p21 and p16		Increase with age, but the only significant difference was between the child group and the elderly group with respect to p21	

## Data Availability

Not applicable.

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
