# Peer review of "Chronological Age Affects MSC Senescence In Vitro—A Systematic Review"

_ijms, 2021, doi:10.3390/ijms22157945_

Round 1

Reviewer 1 Report

I am grateful for the opportunity to review the manuscript entitled "Chronological age affects MSC senescence in vitro - A systematic review" submitted to the International Journal of Molecular Sciences. Within the present study authors applied rather interesting approach to reveal the correlation between donor age and the severity of MSC senescence. At the first glance the answer to this question may seem intuitively obvious – the older the donor the more “senescence-prone” his/her MSC. In fact, this thesis lacks sufficient verification. Within this systematic review the authors performed comprehensive analysis of the existing data on MSC senescence and chronological age. Though the literary findings are rather heterogeneous (different age subgrouping, various senescence markers assessed), summation of the results based on inclusion/exclusion criteria applied by the authors allowed to make reliable and unambiguous conclusion that increased chronological age is associated with increased level of MSC senescence. This observation raises further questions, e.g. what are the causes for enhanced MSC senescence during aging; whether age-related decline in the immune system and/or overall epigenetic erosion accompanying organismal aging could be the underlying causes for elevated MSC senescence; whether application of senolytics/senostatics prior isolation of MSC from aged donors could be a promising strategy for regenerative medicine. To sum up, the present study is worth publishing, since it defines the validity for further detailed investigations in this field, which might be useful for clinics.

There are several minor points that should be revised, e.g. the word “however” is used twice within one sentence (line 15-16).   

Author Response

Please find our revision letter attached.

Reviewer 2 Report

Congratulations to authors for compiling important update on issue pertaining to age vs cellular senescence. I would suggest to expand the discussion by adding clinically important pargraph on genotoxic stress induced senescence (e.g., through changes in senescence messaging secretome) more persistent and difficult to reverse in seniors.

Author Response

(The authors gave the same response as above.)

Reviewer 3 Report

Comments on Kapetanos et al.

In this review, the authors perform a meta-analysis of nine studies that examine the correlation between donor age and MSC characteristics and functions. The analysis and results are very well presented and clear, although the conclusions are a bit ambiguous. This review will be an important contribution to everyone who uses or wishes to learn about hMSC. I would like to note only two minor things I'd like to see added:

  • First, the word human should be added to keywords.
  • Second, given the large number of details given in the results, I think it could be appropriate if the author tries to draw a hypothetical model illustrating the difference between MSC from young vs old donors. Additionally, many cellular factors are mentioned – p53, MAPK, etc. – so It could be nice to have a signal transduction figure that illustrates the effect they have on each other in young and old MSCs. Last, although the author makes clear the distinction between in vivo and in vitro aging, it could be nice to note- in the discussion – if the effects seen in MSC from old donors are somewhat similar to those observed in high passage MSC from a similar source.

Author Response

(The authors gave the same response as above.)

Reviewer 4 Report

In this review, the authors investigated the effects of chronological donor ageing on the senescence of MSCs. They identified 3,023 studies after searching four databases but only nine studies met the inclusion and exclusion criteria and so were included in the final analyses. These studies showed an increase in the expression of p21, p53, p16, ROS, and NF-κB with chronological age, associated with activation of DNA damaging response (DDR), as well as increased levels of stress and inflammation in the MSCs of older donors. An activation of DDR in cells from older donors are associated with a decrease in the expression of proliferative markers. They, also, found an increase in the levels of SA-β-galactosidase, a specific marker of cellular senescence.

The review as well as not innovative in the aspects dealt with, specific goal is laking, that the authors could consider before publishing.  

I suggest a summary figure could make reading easier for the reader.

The authors had reported in the introduction the definition for mesenchymal stromal/stem cells (MSCs). This is the correct definition proposed by ISCT. In fact, the isolation of MSCs according to ISCT criteria produces heterogeneous, non-clonal cultures of stromal cells containing stem cells with different multipotential properties, committed progenitors, and differentiated cells (see recent literature:PMID: 26423725; PMID: 29721206).
For this reason, the authors should uniform in their manuscript. 

Author Response

(The authors gave the same response as above.)

Round 2

Reviewer 4 Report

The use of the term mesenchymal stem cells is not correct. Mesenchymal cells (MSCs) are not a homogenous population but
comprehend several cell types, such as stem cells, progenitor cells, fibroblasts, and other types of cells. For this reason, the best nomenclature is mesenchymal stromal cells

Author Response

Please see cover letter attached
